# Computational Imaging in Dual-Band Infrared Hybrid Optical System with Wide Temperature Range

**DOI:** 10.3390/s22145291

**Published:** 2022-07-15

**Authors:** Shan Mao, Huaile Nie, Tao Lai, Na Xie

**Affiliations:** 1Key Laboratory of Light Field Manipulation and Information Acquisition, Ministry of Industry and Information Technology, and Shaanxi Key Laboratory of Optical Information Technology, School of Physical Science and Technology, Northwestern Polytechnical University, Xi’an 710129, China; niehuaile@163.com (H.N.); lshh322@163.com (T.L.); 2Science and Technology on Electro-Optical Information Security Control Laboratory, Tianjin 300308, China; flourish1994@163.com

**Keywords:** optical design, dual-infrared waveband, diffractive optical element, hybrid optical system design, computational imaging

## Abstract

The special dispersion and temperature characteristics of diffractive optical element (DOE) make them widely used in optical systems that require both athermalization and achromatic aberrations designs. The multi-layer DOE (MLDOE) can improve the diffraction efficiency of the overall broad waveband, but its diffraction efficiency decreases with changes in ambient temperature. When the ambient temperature changes, the micro-structure heights of MLDOE and the refractive index of the substrate materials change, ultimately affecting its diffraction efficiency, and, further, the optical transform function (OTF). In this paper, the influence of ambient temperature on the diffraction efficiency of MLDOE in a dual-infrared waveband is proposed and discussed, the diffraction efficiency of MLDOE caused by ambient temperature is derived, and a computational imaging method that combines optical design and image restoration is proposed. Finally, a dual-infrared waveband infrared optical system with athermalization and achromatic aberrations corrected based on computational imaging method is designed. Results show that this method can effectively reduce the diffraction efficiency of MLDOE by ambient temperature and improve the imaging quality of hybrid optical systems.

## 1. Introduction

Infrared optical systems have the characteristics of good concealment, strong anti-interference ability and, to a certain extent, can identify camouflage targets, which has a good application prospect in space optical antennas [1,2,3,4]. Mid- and long-infrared imaging systems can observe different target types in one optical system, obtaining more target information than only a single waveband. The system has outstanding advantages in identifying authenticity, anti-stealth, and multi-target tracking; it is a high-end development trend of infrared detection, especially for space optical antennas since it has stricter restrictions and requirements for its optical system such as weight and volume, etc. However, there are few infrared optical materials that can be used for both mid- and long-wavebands as it is difficult to design an imaging system covering both the wavebands covering 3–5 and 8–12 μm at the same time. So, the emergence of diffractive optical elements (DOE) provides a new solution, which can show the characteristics of both special dispersion and temperature [5]. Diffractive-refractive hybrid imaging optical systems (D-RHIOS) composed of DOEs and refractive lenses, can be applied to simplify the system, correcting the chromatic aberration, secondary spectrum and athermalization. It is of great significance for hybrid imaging optical system optimization, such as realizing a better image quality, miniaturizing the volume and weight, and avoiding the use of special optical materials [6,7]. Therefore, it has been widely used in military and aerospace optical systems.

For DOE, diffraction efficiency is the very important parameter that determines the working waveband and application. Specifically, MLDOE can obtain high diffraction efficiency overall in a wide waveband, as well as with special dispersion and temperature characteristics [8]. What is more, it can also be applied for dual or more wavebands imaging at the same time. However, diffraction efficiency of MLDOE can be affected by ambient temperature, thereby the imaging quality of D-RHIOS will be affected, leading to the image blurring.

Above all, we propose a design of computational imaging in D-RHIOS, which is solved by combining image restoration model and imaging optical system. Based on scalar diffraction theory, the imaging MLDOE is designed and applied in dual-infrared waveband imaging optical system. Then, an athermalization and achromatic aberrations corrected dual-infrared waveband imaging optical system is designed and to verify the feasibility of the theoretical study. The result shows that this method can reduce the influence of the ambient temperature on the PSF model and can significantly improve the influence of diffraction efficiency reduction in the diffraction computational imaging system. The method and results provide a feasibly theoretical basis for the design of multi-band optical antennas, which can be applied to photoelectric systems such as wide-spectrum imaging, wide-spectrum communications, and astronomical observations and other antenna opt-electronic systems.

## 2. Basic Principle and Method

### 2.1. MLDOE Design in Dual-Infrared Waveband

The MLDOE consists of different optical materials with different dispersion characteristics, and the separated double-layer DOE (DLDOE) is the mostly common structure of MLDOE. It consists of two single layer DOEs, which are closely stacked together with a thin air gap between them, in which the complex amplitude transmittance functions are *t*_1_(*x*, *y*) and *t*_2_(*x*, *y*), respectively. As shown in Figure 1, the incoming and outgoing fields for each layer are ${\tilde{U}}_{1i}$, ${\tilde{U}}_{1t}$ and ${\tilde{U}}_{2i}$, ${\tilde{U}}_{2t}$ for each layer, separately.

Since the two layers are closely attached, it can be considered that the outgoing field ${\tilde{U}}_{1t}$ of the first layer for DLDOE does not propagate through the space, and directly becomes the incoming field ${\tilde{U}}_{2i}$ of the second layer, as ${\tilde{U}}_{1t}={\tilde{U}}_{2i}$. Based on the scalar diffraction theory, the total transmittance of the DLDOE can be obtained as [9]
(1)$$\tilde{t}(x,y)=\frac{{\tilde{U}}_{2t}}{{\tilde{U}}_{1i}}=\frac{{\tilde{U}}_{2t}}{{\tilde{U}}_{2i}}\cdot \frac{{\tilde{U}}_{1t}}{{\tilde{U}}_{1i}}={\tilde{t}}_{2}\cdot {\tilde{t}}_{1}$$

From Equation (1), we can see that the DLDOE that are closely attached can be regarded as one DOE, and its equivalent transmittance is equal to the product of the transmittance of each single-layer DOE. Similarly, results can be derived for DOE with more than two layers. Therefore, the DLDOE can replace the traditional single-layer DOE in D-RHIOS.

In some special cases, the ambient temperature of the two substrate materials of the DLDOE is different, but in most cases, processes covering design, manufacture and assemble are always at the same ambient temperature. Considering the influence of ambient temperature on diffractive micro-structure heights is much greater than that of the longitudinal dimensions, the effects of the diffractive micro-structure heights on its diffraction efficiency caused by ambient temperature is more obvious than that of the longitudinal dimension changes [10]. The micro-structure height for each layer of DLDOE is ten times or more that of the traditional single-layer DOE, so the micro-structure heights of the DLDOE and the refractive index of the base material are changing with ambient environment, which will significantly affect its diffraction efficiency, resulting in the inability to achieve high-quality imaging at different ambient temperatures. When the ambient operating temperature changes, the surface profile of the DLDOE changes, as shown in Figure 2.

In Figure 2, the red area in the left indicates that when the ambient temperature is higher than the designed temperature, the micro-structure height for the first layer of DLDOE expands from the theoretical value *H*_01_ to*H_a_*_1_, and the second layer of DLDOE expands from the theoretical value *H*_02_ to*H_a_*_2_. Similarly, the blue area in the right indicates that when the ambient temperature is lower than the designed temperature, the micro-structure height of the first layer of DLDOE shrinks from the theoretical value *H*_01_ to *H*′*_a_*_1_, and the second layer of DLDOE shrinks from the theoretical value *H*_02_ to *H*′*_a_*_2_.

Here, in order to discuss the effects on diffraction efficiency caused by ambient temperature changes of the DLDOE, the effects of the manufacturing errors and the incident angle are not considered. In addition, it is assumed that the ambient temperature of DLDOE changes slowly, regardless of the temperature gradient effects. Then, based on the scalar diffraction efficiency, phase delay for DLDOE is expressed as [11]
(2)$$\phi_{i}=\frac{2\pi }{\lambda }\sum_{k=1}^{2}H_{0k}(n_{k}-n_{0})$$
where *Φ**_i_* is the designed phase delay, *H*_0*k*_ is the designed micro-structure height of the *k*th layer of DLDOE, *n_k_* is the refractive index of the substrate material of the *k*th layer of DLDOE, *n*_0_ is the refractive index of the medium between the two layers, usually it is with air gap, as *n*_0_ = 1.

After some derivations, we can get the real phase delay caused by ambient temperature, expressed as
(3)$$\phi_{a}=\frac{2\pi }{\lambda }\sum_{k=1}^{2}\left[H_{ak}(n_{k}-n_{0})+H_{0k}\left(\frac{dn_{k}}{dt}-\frac{dn_{0}}{dt}\right)\Delta t\right]$$
where, $\frac{dn_{k}}{dt}$ represents the temperature coefficient of refractive index of the substrate materials corresponding to the *k*th layer of DLDOE, and $\frac{dn_{0}}{dt}$ represents the temperature coefficient of refractive index of air. When the ambient temperature changes slightly, the heights of the micro-structure of the *k*th layer can also changes. The ratio $\alpha_{gk}$ of this change to the design value of the micro-structure heights is called the linear expansion coefficient, which means the change of the *k*th layer of DLDOE in the ambient temperature.

When we use DLDOE in dual-infrared waveband, the optimal design of the bandwidth-integrated average diffraction efficiency (BIADE) of dual-band DLDOE in different wavebands is *ω**_i_* = 0.5, achieved by using the bandwidth integral average diffraction efficiency (BIADE) weight factor of different wavebands. The BIADE of the dual-band DLDOE is [12]
(4)$${\bar{\eta }}_{m-\mathrm{int}}(\lambda_{1},\lambda_{2})=\omega_{1}\frac{1}{\lambda_{\max1}-\lambda_{\min1}}\int_{\lambda_{\min1}}^{\lambda_{\max1}}\eta_{m-1}(\lambda_{1})d\lambda_{1}+\omega_{2}\frac{1}{\lambda_{\max2}-\lambda_{\min2}}\int_{\lambda_{\min2}}^{\lambda_{\max2}}\eta_{m-2}(\lambda_{2})d\lambda_{2}$$

In Equation (4), $\omega_{1}+\omega_{2}=1$, and *ω*_1_ and *ω*_2_ are the weighting factors of the BIADE of the two wavebands, respectively, $\lambda_{\min1}$, $\lambda_{\max1}$ and $\lambda_{\min2}$, $\lambda_{\max2}$ represent the minimum and maximum wavelengths of the two wavebands, respectively. Using the BIADE weighting factors of different wavebands, the intrinsic relationship between the design wavelength and the BIADE of the dual-band DLDOE is established, and the designed micro-structure heights can be calculated to ensure the maximum diffraction efficiency.

What’s more, for D-RHIOS, the real optical transform function (OTF) is affected by the BIADE, expressed as [13]
(5)$$OTF(f_{x},f_{y})=\eta_{Int}\cdot OTF_{m}^{\%}(f_{x},f_{y})+(1-\eta_{Int})\cdot \delta (f_{x})\delta (f_{y})$$
where, $OTF_{m}^{\%}(f_{x},f_{y})$ stands for the OTF when the diffraction efficiency reaches 100%, and it can be obtained after optimization form the design software such as ZEMAX or CODE V. For Equation (5), we can see that the OTF of D-RHIOS is the function of both diffraction efficiency and OTF in theory. What’s more, compared with the accurate result, the approximate design results and accurate design results differ greatly in the low-frequency, and the high-frequency is very close.

When the micro-structure height calculated, caused by ambient temperature, the real micro-structure height $H_{ak}$ is
(6)$$H_{ak}=H_{0k}(1+\alpha_{gk}\Delta t)$$

According to Equation (6), the diffraction efficiency of the first diffraction order can be expressed as
(7)$$\eta_{m}={\sin c}^{2}\left(m-\frac{\phi_{i}}{2\pi }\right)$$

When the ambient temperature changes, the actual diffraction efficiency expression of the DLDOE is obtained as
(8)$$\eta_{a}^{m}=\sin c^{2}\left\{m-\frac{1}{\lambda }\sum_{k=1}^{2}\left[H_{ak}(n_{k}-n_{0})+H_{0k}\left(\frac{dn_{k}}{dt}-\frac{dn_{0}}{dt}\right)\Delta t\right]\right\}$$

The BIADE for DLDOE directly affects the OTF of D-RHIOS. When the ambient temperature changes, as the diffraction efficiency of the DLDOE changes, the BIADE of the corresponding waveband changes at the same time, so that the imaging quality of the system deviates from the design value. In the design waveband, the actual BIADE after the ambient temperature changes is expressed as
(9)$${\bar{\eta }}_{a}^{m}=\frac{1}{\lambda_{\max}-\lambda_{\min}}\int_{\lambda_{\min}}^{\lambda_{\max}}\sin c^{2}\left(m-\frac{\phi_{a}}{2\pi }\right)d\lambda$$

From Equations (7) and (8), the actual diffraction efficiency and BIADE for DLDOE can be calculated under different working wavebands and ambient temperatures. This is of great significance for the image quality evaluation of the D-RHIOS using DLDOE at different ambient temperatures.

### 2.2. Computational Imaging in D-RHIOS

After the optical system is optimized, we can perform the image simulation in Optic Studio software. The object can be regarded as a collection of many point lights, all of which are in the linear superposition set of the light intensity distribution formed by the point spread function (PSF) through the imaging optical system. So, to describe the imaging process, a set of point light sources in the object and the point diffusion function of the system to do convolution is presented, expressed as [14]
(10)$$g(x,y)=x_{PSF}(x,y)⊗f(x,y)+\eta (x,y)$$
where *g*(*x*,*y*) is the simulation plot, *f*(*x*,*y*) is the light source bitmap, *x_psf_*(*x*,*y*) is the degenerate function, and *η*(*x*,*y*) is noise. Since the simulation graph *g*(*x*,*y*) is obtained by convolving the light source bitmap *f*(*x*,*y*) with the degradation function of the system, *x_psf_*(*x*,*y*), the effect of noise is not taken into account here.

Processing for computational imaging is: Enter a sharp grayscale image as a true infrared image from natural scene, light source bitmap, simulation map, and PSF grid diagram can be obtained respectively after the D-RHIOS optimized; the simulation diagram here is derived regarding to the diffraction efficiency of DLDOE; and finally, images are output separately from a heavy structure with large aberrations, and the simulation effect can be obtained by comparing the difference between the simulation map and the light source bitmap, as shown in Figure 3.

## 3. Design and Analysis

In order to verify the design and advantages of the computational imaging for dual-infrared waveband optical system, based on DLDOE, we design an athermalization and achromatic aberrations corrected dual-infrared waveband optical system. The design ambient temperature range is −40–60 °C, the focal length is 100 mm, the *F* number is 2, and the full field of view is 4°. The detector is a dual-band refrigeration detector with a size of 320 × 256 pixels and a size of 30 μm. The specific parameters are shown in Table 1.

### 3.1. Dual-Band Hybrid Optical System Design

Considering the advantages of optical passive athermalization [15], we use the thermal properties of different optical materials and mechanical structural materials in optical system to compensate for each other and offset the effect of temperature on it.

Here, since the design adopts a one-time imaging scheme, and the initial structure adopts a Petzval-type structure, which is not sensitive to the temperature changes and is easier to achieve athermalization [16]. A DLDOE is added on the two asperical surfaces, and the special chromatic aberration and thermal characteristics of the element are used together with the refractive element to correct the chromatic aberration and thermal aberration of the system. After optimization, the system consists of a total of six asperical lenses, which are made of germanium, AMTIR1, zinc sulfide and zinc selenide. The slicing board is located at the rear end of the system to achieve 100% cold diaphragm efficiency. The total length of the system is 150.8 mm, and its parameters are shown in Table 2 and Table 3. Layout of the optimized D-RHIOS is shown in Figure 4.

The profile of the DOE is expressed as
(11)$$\varphi (r)=\frac{2\pi }{\lambda }\left(A_{1}r^{2}+A_{2}r^{4}+\cdots \right),$$
where, *r* stands for the normal radius, *A*_1_ and *A*_2_ are the coefficients optimized from the ZEMAX software.

After optimization, its MTF is shown in Figure 5, at 20 °C, the MTFs at the cut-off frequency is greater than 0.78 and 0.55 in mid- and long-infrared waveband; at −40 °C, the MTFs at the cut-off frequency is greater than 0.49 and 0.48 in mid- and long-infrared waveband; at 60 °C, the MTFs at the cut-off frequency is greater than 0.78 and 0.55 in mid- and long-infrared waveband; at 60 °C, the MTFs at the cut-off frequency is greater than 0.70 and 0.58 in mid- and long-infrared waveband. In addition, this design does not consider the diffraction efficiency, all wavelengths of MTF in the range of −40–60 °C, indicating that this dual-band infrared optical system can be more clearly imaged.

### 3.2. DLDOE Design

In this design, the substrate materials of DLDO are ZNSE and GE, respectively, which are capable of being susceptible to diffraction efficiency in dual-infrared waveband, so it needs to be optimally designed and computationally imaged [17,18,19]. The DLDOE overcomes the disadvantage that the diffraction efficiency of single-layer DOE deviates from the design wavelength and has been applied in both visible and infrared broad- or multi- band optical systems. The effect of ambient temperature on the diffraction efficiency when the DLDOE is applied to different working bands will be discussed in detail below. In the infrared band, ZNSE and GE are selected as the substrate materials of the DLDOE. The temperature parameters of the above substrate materials are listed in Table 4.

In addition, design results of the DLDOE are shown in Table 5.

In dual-infrared waveband, ZNSE and GE are used as the substrate materials of the DLDOE, the relationship between the actual diffraction efficiency and the operating wavelength and ambient temperature is shown in Figure 6. Relationship between ambient temperature and BIADE is shown in Figure 7, and the BIADE at different ambient temperatures is listed in Table 6.

From Figure 6 and Figure 7, it can be seen that compared with diffraction efficiency in long-infrared waveband, the mid-infrared waveband is more susceptible to the effects of ambient temperature, resulting in a decrease in diffraction efficiency and the BIADE, so the algorithm and weight factor used in the subsequent processing of the infrared images will be different.

From the above analysis, it can be seen that the diffraction efficiency of DLDOE composed of ZNSE and GE is very sensitive to the change of ambient temperature, so the MTF for D-RHIOS with DLDOE combing ZNSE and GE will image and should be restored when the ambient temperature changes. The quality is obviously degraded, and subsequent computational imaging is required to obtain an image that meets the imaging requirements.

### 3.3. Computational Athermalization Image Restoration and Its Evaluation

For mid- and long-infrared wavebands imaging, the MTF obtained from the optical design software does not consider the influence of diffraction efficiency. However, in fact, its results on imaging for the detector should consider the effects of diffraction efficiency reduction. So, we applied an algorithm for imaging restoration and evaluation. The processing is as follows:

(1)Figure 8 shows the real and blurred images under different ambient temperatures in the long- and mid-infrared wavebands obtained by image simulation from D-RHIOS. It can be seen that the low diffraction efficiency leads to obvious image blurring;(2)The degradation function of the system, *x_psf_*(*x*,*y*), is spatially variable, because the field of view angle of the system is 2°, which can be approximated as a spatially unchanged system within a small angle range. Therefore, image restoration can be carried out in a spatially unchanging deconvolution method;(3)The blind deconvblind function in MATLAB is used to blur the image. The PSF of the central field of view is taken as an estimate of the degenerative function *xpsf*(*x*,*y*), and it is substituted with the simulation diagram *g*(*x*,*y*) into the deconvblind function, and the restored image and the restored point diffusion function PSF can be obtained;(4)The edge of the restored image obtained by this method is seriously distorted, so when importing the image, the image that is slightly larger than the system design field of view can be imported, and the corresponding recovered image is obtained. The light source bitmap, simulation map, and recovery image are all cropped accordingly, and the severe distortion of the edge is removed before comparison;(5)The image used this time has a field of view height of slightly greater than 2° in the center after cropping, and the image resolution is 961 × 721. The pixel size of the analog detector is required to be 1024 × 768 and the pixel size is 5 microns.

Comparing the restored image with the simulated image, it can be subjectively seen that the image quality has been significantly improved. From Figure 9, it can also be seen that there is a significant improvement.

**Figure 8 sensors-22-05291-f008:**
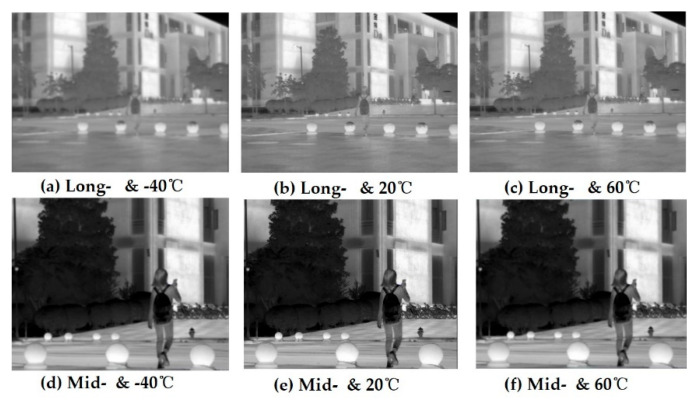
The original imaging of simulation considering diffraction efficiency effects.

## 4. Conclusions

Considering that diffraction efficiency is sensitive to ambient temperature, and can cause MTF reduction for a real D-RHIOS, we have proposed a computational design for dual-infrared wavebands for temperature and color aberration corrections for D-RHIOS in case of diffraction efficiency and present this design in detail. The results show that DLDOE can be applied in dual-infrared wavebands for aberration corrections and is more practical for optical engineering, especially for D-RHIOS with DOE, and can be used, significantly, for quantization and optimization in practically engineered D-RHIOS.

## Figures and Tables

**Figure 1 sensors-22-05291-f001:**
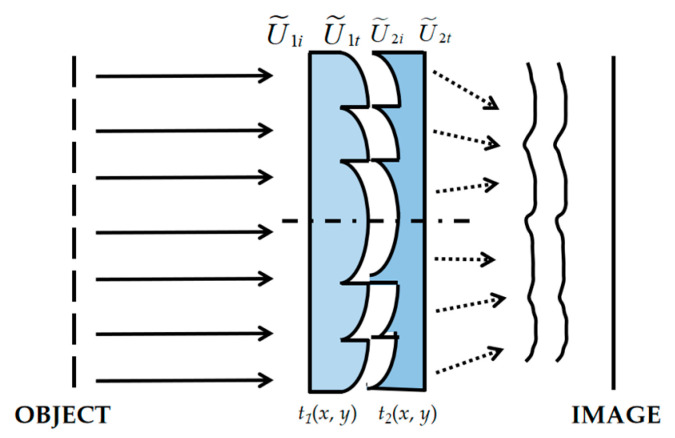
The incoming and outgoing fields for DLDOE.

**Figure 2 sensors-22-05291-f002:**
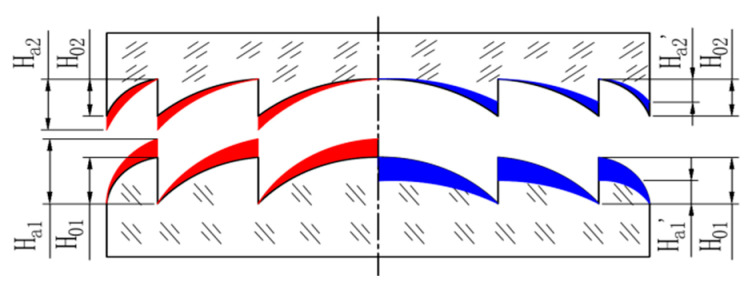
Surface profile of DLDOE when ambient temperature changes.

**Figure 3 sensors-22-05291-f003:**
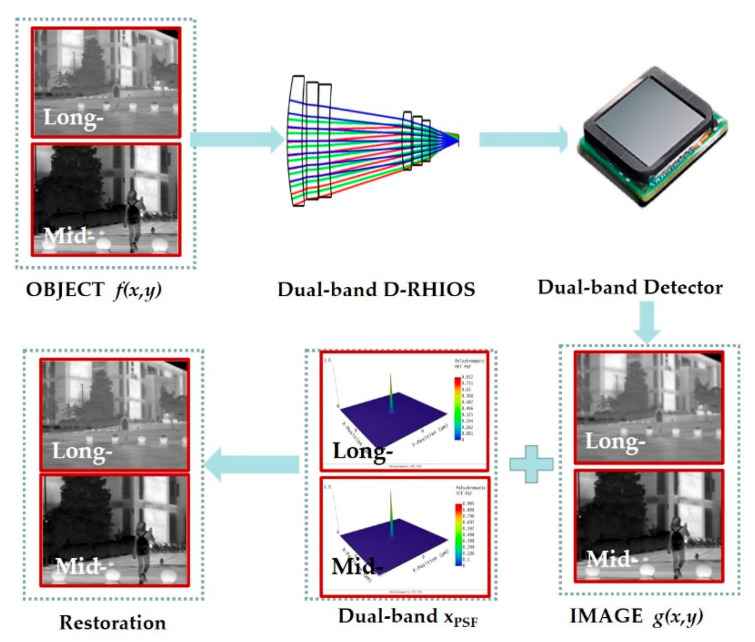
Diagram of the schematic.

**Figure 4 sensors-22-05291-f004:**
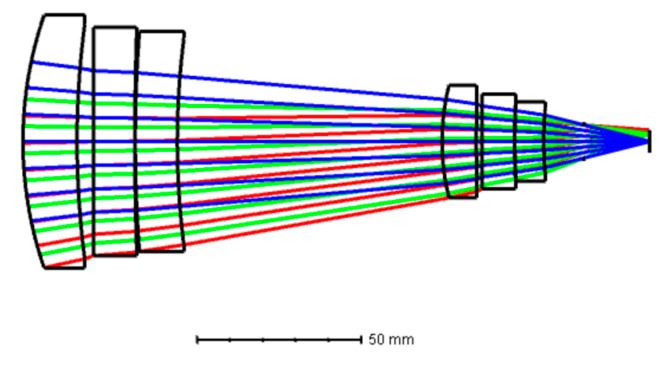
Layout of the dual-infrared waveband imaging optical system.

**Figure 5 sensors-22-05291-f005:**
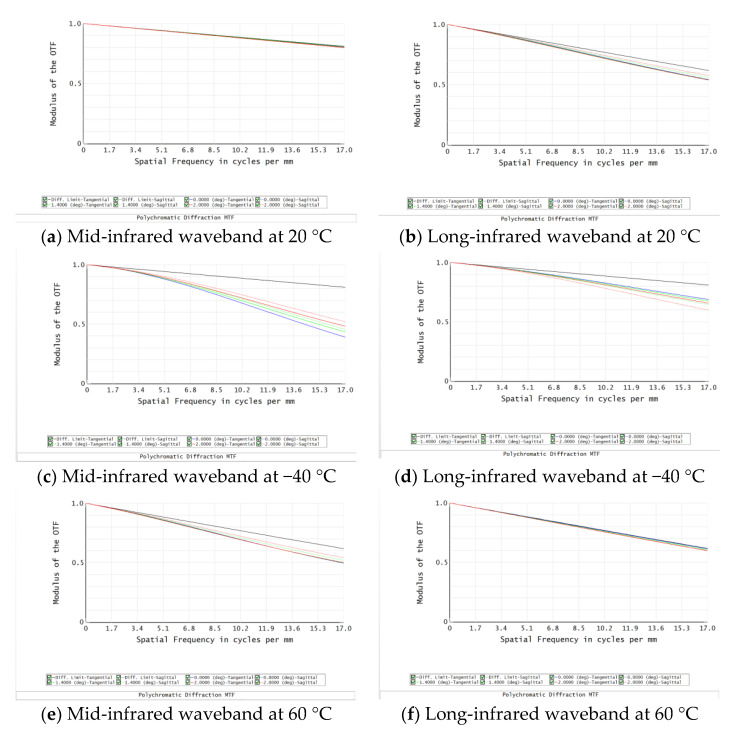
MTF for D-RHIOS at different ambient temperatures.

**Figure 6 sensors-22-05291-f006:**
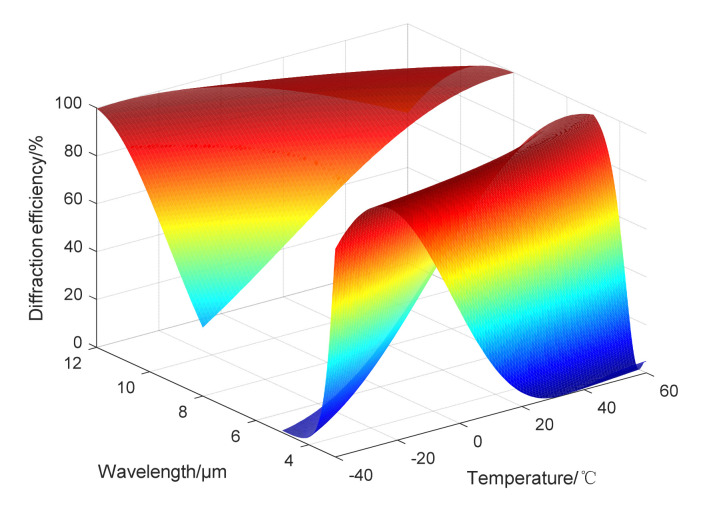
Relationship among diffraction efficiency-waveband-ambient temperature for DLDOE in dual-infrared waveband.

**Figure 7 sensors-22-05291-f007:**
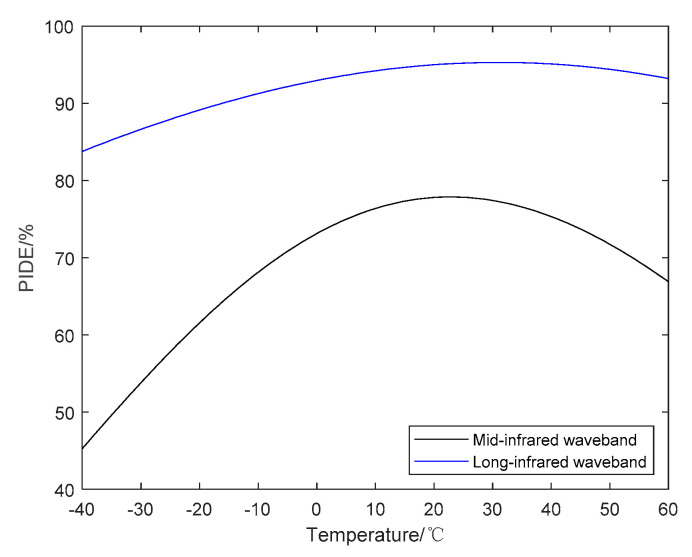
Relationship between BIADE and ambient temperature for DLDOE in dual-infrared waveband.

**Figure 9 sensors-22-05291-f009:**
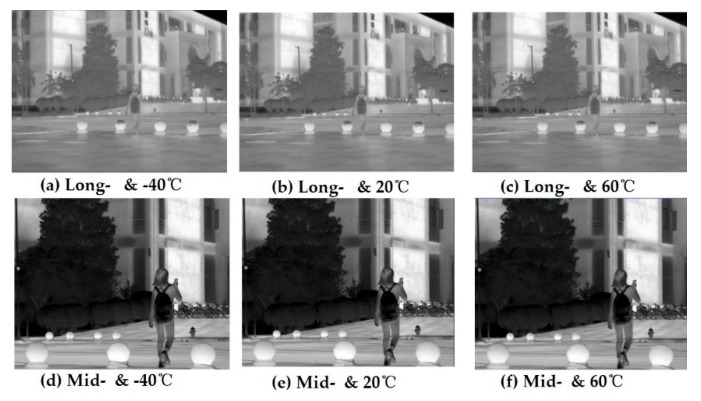
The optimal imaging after restoration.

**Table 1 sensors-22-05291-t001:** Design specifications of D-RHIOS.

	Paremeter	Value
Imaging optical system	Waveband/μm	3~5 and 8~12
	Field of view/°	10
	Effective focal length/mm	100
	Aperture/mm	50
Ambient temperature	Temperature Range/°C	−40~60
Detector	Pixel/μm	30
	Image size	320 × 256
Lens barrel material	AL	$\alpha_{b}=23.6\times 10^{-6}$

**Table 2 sensors-22-05291-t002:** Layout of D-RHIOS.

Surface	Type	Radius (mm)	Thickness (mm)	Materials	$TCE\times 10^{-6}$
Object	Standard	Infinity	Infinity	--	--
1	Standard	117.744	16.481	AMTIR1	--
2	Standard	272.931	4.970	--	23.6
3	Standard	9.793 × 10^5^	12.650	GE	--
4	Standard	1027.119	0.1	--	23.6
5	Standard	441.957	12.819	ZNS	--
6	Standard	247.541	80.641	--	23.6
7	Standard	69.898	10.444	ZNSE	--
8	Standard	998.210	1.672	--	23.6
9	Standard	2194.902	9.999	ZNSE	--
10	Binary	239.218	0.044	--	23.6
11	Binary	239.218	8.820	GE	--
12	Standard	111.391	12.276	--	23.6
Stop	Standard	Infinity	19.995	--	23.6
Image	Standard	Infinity	--	--	--

**Table 3 sensors-22-05291-t003:** Surface profile of the DLDOE.

Surface	Normal Radius	A_1_	A_2_
10	100	−1.4227 × 10^3^	1.0222 × 10^4^
11	100	−1.4227 × 10^3^	1.0222 × 10^4^

**Table 4 sensors-22-05291-t004:** Temperature parameters of substrate materials in different wavebands.

Waveband	MWIR		LWIR	
Substrate Materials	GE	ZNSE	GE	ZNSE
*α* (×10^−6^/°C)	5.7	7.1	5.7	7.1
*dn*/*dt* (×10^−6^/°C)	424	63	404	61

**Table 5 sensors-22-05291-t005:** Design results of the DLDOE.

Wavebands/µm	3~5	8~12
Design wavelength/µm	3.8	10.2
Design order/%	−137 and 138	−50 and 51
*H*_1_/µm	365.013	
*H*_2_/µm	174.133	

**Table 6 sensors-22-05291-t006:** BIADE for DLDOE at different ambient temperatures in dual bands.

Ambient Temperature/°C	BIADE/%	
	MWIR	LWIR
−40	45.289	83.773
20	77.799	94.990
60	66.880	93.190

## Data Availability

Not applicable.

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
