# Peer review of "Computational Imaging in Dual-Band Infrared Hybrid Optical System with Wide Temperature Range"

_sensors, 2022, doi:10.3390/s22145291_

Round 1

Reviewer 1 Report

This paper deals with a new methods of processing thermal imagesand is of great interest as one of the innovative image processing. It would be worthy of publication if the following points were corrected.

Infrared radiation is distributed over a wide range of wavelengths, from 830 nm to about 1 mm. What extent is the method proposed by the authors applicable to this wavelength range?

 Various methods are used for thermal imaging. Please state what advantages the author's method has over conventional methods.

 It is well written. I consider it worthy of acceptance if the sections I commented on to the author can be corrected.

Author Response

We thank you very much for your kind efforts and encouragements on our manuscript titled “Computational imaging in dual-band infrared hybrid optical system with wide temperature range” (Manuscript ID: sensors-1771600).

  And we truly appreciate you giving us this opportunity to revise our manuscript to be more perfect. We have responded to every comment from the Reviewers in detail in the following responses and revised this manuscript accordingly (please see our changes in the manuscript underlined). Besides, we have re-checked our manuscript on words, sentence grammars, formulas, figures, tables and the logicality to ensure the validity and accuracy. We hope that this revision is acceptable and look forward to hearing from you soon.

This paper deals with a new methods of processing thermal images and is of great interest as one of the innovative image processing. It would be worthy of publication if the following points were corrected.

Infrared radiation is distributed over a wide range of wavelengths, from 830 nm to about 1 mm. What extent is the method proposed by the authors applicable to this wavelength range?

Various methods are used for thermal imaging. Please state what advantages the author's method has over conventional methods.

It is well written. I consider it worthy of acceptance if the sections I commented on to the author can be corrected.

Response: Thank you very much for your kind comments and professional questions.

(1) Indeed, infrared radiation is distributed over a wide range of waveband, almost from 830 nm to 1 mm. The working waveband of the refractive-diffractive hybrid imaging optical system (D-RHIOS) designed in our manuscript includes the mid- and long- infrared wavebands, that is, it can simultaneously achieve the dual-infrared waveband simultaneous imaging with a common aperture covering both 3-5µm and 8-12µm wavebands. Since diffraction efficiency can be reduced by wavelengths and ambient temperature, method proposed in this manuscript can be realized as long as the imaging waveband conformed to the detection range of the detectors as well as diffraction efficiency requirement affected by wavebands and ambient temperature. 

(2) There are many thermal imaging methods, mainly including active imaging and passive imaging, whose difference is whether there is an external infrared light source. There are also various structures of thermal imaging optical systems, such as reflective, refractive, diffractive types and so on. For multi-band imaging optical system structures, including multi-aperture and common aperture, the dual-band infrared optical system in our manuscript is a passive common-aperture infrared imaging optical system, which can minimize the size, reduce the weight, reduce the manufacture difficulty, and reduce the cost of the thermal imaging optical system.

For the above two professional comments, we have described them in detail in the revised manuscript, which has been marked in red.

Last but not least, we have modified the whole manuscript according to your valuable suggestions and comments, which are very helpful to substantially improve the quality of this manuscript. Thanks again for the careful and professional works.

Reviewer 2 Report

Review of the manuscript entitled “Computational imaging in dual-band infrared hybrid optical system with wide temperature range” by Shan Mao et al.

The manuscript explains the application of a computational method to improve blurred images. The images are taken with supposedly infrared wavelengths 3-5, 8-12 microns.

Some questions arise when reading the manuscript and some suggestions are given to clarify some aspects. Probably you (authors) will send me a text with your response but the main objective is that YOU SHOULD BE CLEAR IN THE TEXT OF YOUR MANUSCRIPT.

1 In the future this theoretical work should be tested with a real application. Give advice to the researchers that will perform the experiments.

2 In Fig. 2 is shown a Diffractive Optical Element. It is supposed that a DOE has some high frequency relieves. However, in the examples shown in section 3 there are not high frequency elements. Then why you apply the theory of DOE.

3 In Fig. 8 and 9 you considered an image that was not taken with an infrared detector. Why you did not get one of such images.

4 Paragraph beginning in line 103 describes the phenomena of the height changing profile with temperature. But does not consider the changing in the longitudinal scale. Why?.

5 Line 130. It is mentioned the technique BIADE. This technique is important to understand the manuscript so make a description of it, do not just mention the technique and give reference.

6 What is the distance between the elements in the double layer of Fig. 1 that is considered good to accept that the outgoing field of the first layer becomes the incoming field of second layer?. What are the lenses in Fig. 4 that have this minimum distance?.

7 In section 3.1 an example is shown. A total of 6 aspherical surfaces are mentioned. It also says that a “DLDOE is added on the two aspherical surfaces.” Fully describe or explain the DLDOE.  

Author Response

We thank you very much for your kind efforts and encouragements on our manuscript titled “Computational imaging in dual-band infrared hybrid optical system with wide temperature range” (Manuscript ID: sensors-1771600).

  And we truly appreciate you giving us this opportunity to revise our manuscript to be more perfect. We have responded to every comment from the Reviewers in detail in the following responses and revised this manuscript accordingly (please see our changes in the manuscript underlined). Besides, we have re-checked our manuscript on words, sentence grammars, formulas, figures, tables and the logicality to ensure the validity and accuracy. We hope that this revision is acceptable and look forward to hearing from you soon.

Review of the manuscript entitled “Computational imaging in dual-band infrared hybrid optical system with wide temperature range” by Shan Mao et al. The manuscript explains the application of a computational method to improve blurred images. The images are taken with supposedly infrared wavelengths 3-5, 8-12 microns.

Some questions arise when reading the manuscript and some suggestions are given to clarify some aspects. Probably you (authors) will send me a text with your response but the main objective is that YOU SHOULD BE CLEAR IN THE TEXT OF YOUR MANUSCRIPT.

1 In the future this theoretical work should be tested with a real application. Give advice to the researchers that will perform the experiments.

Response: Thank you very much for your kind suggestion. In the current manuscript, we have investigated the theory, demonstrating the correctness and feasibility of this theory. And now, we are making a proto-type machine based on this principle, and it will be applied to the experiments of the actual system in the future.

2 In Fig. 2 is shown a Diffractive Optical Element. It is supposed that a DOE has some high frequency relieves. However, in the examples shown in section 3 there are not high frequency elements. Then why you apply the theory of DOE.

Response: Thank you very much for your professional question. It is indeed that a DOE has some high frequency relieves, however, in our work, we apply the DOE (DLDOE mainly) in imaging optical system, which only bears a small optical power and used for aberration corrections in refractive-diffractive hybrid imaging optical system (D-RHIOS). Therefore, the first-order diffraction order can meet the design requirements. In addition, considering that the feature size of DLDOE in D-RHIOS is much larger than the incident wavelength, the scalar diffraction theory can also meet the design accuracy requirements.

3 In Fig. 8 and 9 you considered an image that was not taken with an infrared detector. Why you did not get one of such images.

Response: Thank you for your very professional question. Figure 8 is the real imaging result of dual-infrared waveband images obtaining from dual-infrared detectors after imaging through D-RHIOS. And Fig. 9 is the restored image results of computation imaging considering diffraction efficiency and image blur restored with the simulated image, it can be subjectively seen that the image quality has been significantly improved, verifying the correctness of the model and method in our manuscript.

4 Paragraph beginning in line 103 describes the phenomena of the height changing profile with temperature. But does not consider the changing in the longitudinal scale. Why?

Response: Thank you for your very professional question. The influence of ambient temperature on diffractive micro-structure heights is much greater than that of the longitudinal dimensions, and the effects of the diffractive micro-structure heights on its diffraction efficiency caused by ambient temperature is more obvious than that of the longitudinal dimension changes. The effect on diffraction efficiency of the longitudinal scale is very small and can be ignored. This has been reported in the corresponding reference (Ref. 10), in addition, we have added a corresponding note in the revised manuscript and incorporated reference.

[10] M. Piao, Q. Cui, H. Zhu, C. Xue, B. Zhang. Diffraction efficiency change of multilayer diffractive optics with environmental temperature. Journal of Optics, 2014, 16(3):035707.

5 Line 130. It is mentioned the technique BIADE. This technique is important to understand the manuscript so make a description of it, do not just mention the technique and give reference.

Response: Thank you very much for your kind suggestion. Since the bandwidth integral average diffraction efficiency (BIADE) of the multi-layer diffractive optical element (MLDOE) has a certain effect on the optical transfer function (OTF) of the optical system, it is necessary to optimize the design of the MLDOE in the refractive-diffractive hybrid imaging optical system (R-DHIOS), so when the BIADE of the MLDOE is the maximum value, the OTF can obtain the maximum. In addition, we describe BIADE in more detail in this revised manuscript and include corresponding references.

[12] C. Xue, Q. Cui. Design of multilayer diffractive optical elements with polychromatic integral diffraction efficiency. Opt. Lett. 35(7): 986-988 (2010).

6 What is the distance between the elements in the double layer of Fig. 1 that is considered good to accept that the outgoing field of the first layer becomes the incoming field of second layer? What are the lenses in Fig. 4 that have this minimum distance?

Response: Thank you very much for your professional question. For the separated double-layer diffractive optical element (DLDOE) in our manuscript, the middle gap is an air space, and the air space will affect the angle of incidence to the second-layer DLDOE, and also affect its diffraction efficiency. Therefore, after lots of analysis and calculations, and in practical design and engineering applications, in order to meet the design requirements, the air gap of the DLDOE is generally controlled within 0.05mm.

7 In section 3.1 an example is shown. A total of 6 aspherical surfaces are mentioned. It also says that a “DLDOE is added on the two aspherical surfaces.” Fully describe or explain the DLDOE.  

Response: Thank you very much for your kind comment. This dual-band infrared optical system consists of 6 aspherical lenses and the DLDOE is added on the two aspherical surfaces. We have added Table 3 and the corresponding expressions in the revised manuscript, marked in red.

Last but not least, we have modified the whole manuscript according to your valuable suggestions and comments, which are very helpful to substantially improve the quality of this manuscript. Thanks again for the careful and professional works.

Reviewer 3 Report

In this paper, the influence of ambient temperature on the diffraction efficiency of Multi-layer DOE is discussed. A dual-infrared waveband infrared optical system is designed and a computational athermalization method for image restoration is applied to improve the image quality. However, there are some questions:

1.     Surface 10 and 11 in Table2 are binary type, more information of the above two surfaces should be provided in the text, such as surface profile, etc.

2.     The diffraction efficiency-waveband-ambient temperature as shown in Fig. 6 should be explained in detail, it seems that the diffraction efficiency owns different characteristics in the mid and long infrared waveband.

3.     Fig. 8 and 9 present the imaging and restoration results under different diffraction efficiency, the influence of the diffraction efficiency on the imaging should be included.

4.   In line 112, "is not considered" should be "are not considered".

Author Response

We thank you very much for your kind efforts and encouragements on our manuscript titled “Computational imaging in dual-band infrared hybrid optical system with wide temperature range” (Manuscript ID: sensors-1771600).

  And we truly appreciate you giving us this opportunity to revise our manuscript to be more perfect. We have responded to every comment from the Reviewers in detail in the following responses and revised this manuscript accordingly (please see our changes in the manuscript underlined). Besides, we have re-checked our manuscript on words, sentence grammars, formulas, figures, tables and the logicality to ensure the validity and accuracy. We hope that this revision is acceptable and look forward to hearing from you soon.

In this paper, the influence of ambient temperature on the diffraction efficiency of Multi-layer DOE is discussed. A dual-infrared waveband infrared optical system is designed and a computational athermalization method for image restoration is applied to improve the image quality. However, there are some questions:

  1. Surface 10 and 11 in Table2 are binary type, more information of the above two surfaces should be provided in the text, such as surface profile, etc.

Response: Thank you very much for your professional and kind comment. In the revised manuscript, we have added the parameters of surface profiles in Table 3 and the corresponding expressions in revised manuscript.

  1. The diffraction efficiency-waveband-ambient temperature as shown in Fig. 6 should be explained in detail, it seems that the diffraction efficiency owns different characteristics in the mid and long infrared waveband.

Response: Thank you very much for your kind comment. Diffraction efficiency is affected by both ambient temperature and wavelength. When the waveband and ambient changing, diffraction efficiency will be different. The micro-structure heights for DLDOE are different with the ambient temperature changing, as shown in Eqs. (3) and (4), the micro-structure heights directly affect phase delay function of DLDOE, which in turn affects its diffraction efficiency. In addition, diffraction efficiency of DLDOE is related to the incident wavelength, so it is therefore also affected. In summary, the ambient temperature and incident wavelength will cause changes in the diffraction efficiency of DLDOE, and when the ambient temperature changes, the wavelength will affect its diffraction efficiency. In addition, we have described it in detail in the revised manuscript, which has been marked in red.

  1. Fig. 8 and 9 present the imaging and restoration results under different diffraction efficiency, the influence of the diffraction efficiency on the imaging should be included.

Response: Thank you very much for your kind comment and professional question. For D-RHIOS, the real optical transform function (OTF) is affected by the BIADE, expressed in Eq. (5). Imaging and restoration results in Fig. 9 are taken both diffraction efficiency and main computational imaging technique into consideration, whose details are listed in section 3.3. 

  1. In line 112, "is not considered" should be "are not considered".

Response: Thank you very much for your kind comment and we have corrected "is not considered" into "are not considered" in the corresponding part.

Last but not least, we have modified the whole manuscript according to your valuable suggestions and comments, which are very helpful to substantially improve the quality of this manuscript. Thanks again for the careful and professional works.

Round 2

Reviewer 2 Report

It will be interesting to read the manuscript describing the relation between the theoretical work developed here and the experimental work.

Reviewer 3 Report

The paper has been revised as required, explaining in detail the issues related to diffraction efficiency of optical systems and has been revised in the appropriate places in the text.